# High-Production-Rate Fabrication of Low-Loss Lithium Niobate Electro-Optic Modulators Using Photolithography Assisted Chemo-Mechanical Etching (PLACE)

**DOI:** 10.3390/mi13030378

**Published:** 2022-02-26

**Authors:** Rongbo Wu, Lang Gao, Youting Liang, Yong Zheng, Junxia Zhou, Hongxin Qi, Difeng Yin, Min Wang, Zhiwei Fang, Ya Cheng

**Affiliations:** 1State Key Laboratory of High Field Laser Physics, Shanghai Institute of Optics and Fine Mechanics, Chinese Academy of Sciences, Shanghai 201800, China; rbwu@siom.ac.cn (R.W.); langgao@siom.ac.cn (L.G.); yindf@siom.ac.cn (D.Y.); 2University of Chinese Academy of Sciences, Beijing 100049, China; 3State Key Laboratory of Precision Spectroscopy, East China Normal University, Shanghai 200062, China; 51180920021@stu.ecnu.edu.cn (Y.L.); 52210920032@stu.ecnu.edu.cn (Y.Z.); 52180920026@stu.ecnu.edu.cn (J.Z.); mwang@phy.ecnu.edu.cn (M.W.); zwfang@phy.ecnu.edu.cn (Z.F.); 4XXL—The Extreme Optoelectromechanics Laboratory, School of Physics and Electronics Science, East China Normal University, Shanghai 200241, China; 5Collaborative Innovation Center of Light Manipulations and Applications, Shandong Normal University, Jinan 250358, China; 6Shanghai Research Center for Quantum Sciences, Shanghai 201315, China

**Keywords:** lithium niobate, electro-optic modulator, insertion loss, photolithography assisted chemo-mechanical etching

## Abstract

Integrated thin-film lithium niobate (LN) electro-optic (EO) modulators of broad bandwidth, low insertion loss, low cost and high production rate are essential elements in contemporary interconnection industries and disruptive applications. Here, we demonstrated the design and fabrication of a high performance thin-film LN EO modulator using photolithography assisted chemo-mechanical etching (PLACE) technology. Our device shows a 3-dB bandwidth over 50 GHz, along with a comparable low half wave voltage-length product of 2.16 Vcm and a fiber-to-fiber insertion loss of 2.6 dB. The PLACE technology supports large footprint, high fabrication uniformity, competitive production rate and extreme low device optical loss simultaneously, our result shows promising potential for developing high-performance large-scale low-loss photonic integrated devices.

## 1. Introduction

Integrated Mach–Zehnder modulators of high light-modulation rates, low power consumption and small sizes are essential elements in contemporary interconnection industries. They are also promising candidates to serve as the building blocks for disruptive applications ranging from quantum information processing and microwave photonics to artificial neural networks [1,2,3,4,5,6]. To meet the stringent requirements of industrial standards, including a low optical insertion loss, low drive voltage, broad bandwidth, dense integration, robustness, low cost and high production rates, various material platforms and fabrication approaches have been investigated. For instance, silicon (Si), indium phosphide (InP), polymers and plasmonics have been explored to construct modulators [7,8,9,10] Although these platforms separately offer the advantages of compact footprints (Si, plasmonics) and low drive voltages (InP, polymer), drawbacks including large *V_π_* (Si), high optical loss (plasmonics), nonlinear response (InP), or stability problem (polymer) still exist in modulators based on these material platforms. Notably, thin film lithium niobate (TFLN), which is a relatively new optical material featuring a high electro-optic coefficient, a large optical nonlinearity and a broad transmission window, has attracted significant attention over the past decade [11,12,13,14,15,16,17,18,19,20,21,22,23,24]. Unlike the conventional bulk LN electro-optical modulators widely used nowadays in the optical communication industry, integrated photonic circuits (PICs) built upon the TFLN can readily break up the bottlenecks of the bulk LN modulators in terms of the device size, drive voltage and bandwidth. On the other hand, with the shrinking sizes of the optical mode profiles in the ridge waveguides formed on the TFLN substrate, challenges have been encountered in the high-production-rate fabrication of low-loss, large-scale TFLN PIC devices. Electron beam lithography followed by reactive ion etching technology can provide sufficiently high fabrication precision whilst suffering from a relatively low production yield (more than 8 days for 6-inch wafer exposure [16]). Ultraviolet (UV) lithography technology can provide high fabrication efficiency, whilst uncertainty still exists in terms of uniformity of wafer-scale production and propagation loss (over 0.1 dB/cm) induced by the sidewall roughness [18,21].

To address the challenges, photolithography assisted chemo-mechanical etching (PLACE) technology has emerged as a promising technology for thin-film LN device production. Utilizing the femtosecond direct writing technology for Cr hard mask patterning and followed by chemo-mechanical polishing (CMP) technology for waveguide etching, the PLACE technology supports large device footprint (footprint size over 12 inch), high fabrication uniformity and competitive production rate (taking only ~3 min to mask patterning for each modulator) simultaneously thanks to the high average power femtosecond laser and high-speed large-motion-range position stages. PLACE also provides extreme low device optical loss thanks to the extreme smooth waveguide sidewall (roughness less than 0.1 nm) produced by the CMP process. During the last few years, various micro photonic components fabricated using PLACE technology including a microresonator with quality up to 10^8^ [25], waveguides with propagation loss as low as 0.027 dB/cm [26] and waveguide delay lines with meter scale length [27] have been reported.

Here we demonstrate high performance thin-film LN electro-optic (EO) modulators fabricated using PLACE technology. Our device shows a 3-dB electro-optic bandwidth over 50 GHz, along with a half wave voltage-length product (*V_π_L*) of 2.16 Vcm. We obtain a fiber-to-fiber insertion loss of 2.6 dB.

## 2. Simulations, Device Design and Fabrication

Figure 1a shows the simulated optical field of the fundamental transverse electric (TE) mode for a typical LN ridge waveguide fabricated by PLACE technology using COMSOL Multiphysics (V5.6, Stockholm, Sweden) under perfect matched layer (PML) boundary condition, the geometries used in the simulation were extracted from the SEM images of the cross sections of waveguides fabricated using PLACE technology. The electrical field distribution in the waveguide with a 1 V static voltage applied between the signal and ground electrodes is shown in Figure 1b. To optimize the optical propagation loss caused by the absorption of the electrodes with a reasonable half wave voltage-length production (*V_π_L*), we calculated both the *V_π_L* and the optical propagation loss for different electrode-pair gaps and etching depth, as shown in Figure 1c. Considering the radio-frequency (RF) loss, the alignment accuracy of the lift off process and the fabrication stability of the PLACE process, we choose an electrode-pair gap of 5.5 μm and a waveguide etching depth of 0.21 μm, which corresponds with a theoretical *V_π_L* of 2.2 Vcm and an optical loss below 0.1 dB/cm.

The design of the EO LN modulator is illustrated in Figure 2a. The modulators are manufactured upon a commercial x-cut lithium niobate on isolator (LNOI) wafer (NANOLN) with a thin-film LN thickness of 500 nm, which is bonded to a buried silica (SiO_2_) layer on a 500-μm-thick silicon (Si) substrate. The modulator consists of two waveguide arms connected by a beam splitter and a combiner based on multimode interference (MMI) couplers. The electrodes carrying the RF signals propagating along the optical waveguide arms are designed and fabricated with the ground-signal-ground (GSG) configuration. Two groups of terminal pads with characteristic impedance of 50 Ω are fabricated on both the ends of the coplanar waveguide (CPW) transmission lines. The cross section of the modulator is shown in Figure 2b. The GSG electrodes are arranged on the same plane of the waveguide arms to achieve better EO interaction. A silicon oxynitride (SiON) layer is deposited on top of the fabricated waveguides and the electrodes to obtain a better matching between the group velocity of optical field and RF field. Meanwhile, the SiON layer also functions as the cladding waveguide for the LN inverse tapers fabricated at both the ends of the modulator which together form the spot size converter (SSC). The SSC provides high optical coupling efficiency between the fabricated modulator and ultra-high numerical aperture fibers, giving rise to a favorable low fiber-to-fiber optical insertion loss.

Based on the geometric configuration in Figure 1, the group velocity matching between the RF field on the travelling electrode and the TE mode optic field in the ridge waveguide is numerically analyzed using high frequency structure simulator (HFSS) software (ANSYS Electronics Suite 2021 R2, Ansys Inc., Canonsburg, PA, USA), under absorbing condition. We plot the simulation results in Figure 2c. As shown by the curve of group refractive index versus frequency, the group velocity of the RF signals matches nicely with that of the optical field (mismatching below 0.02 when RF frequency from 40 to 70 GHz). Furthermore, we present the simulated characteristic impedance of the travelling electrode in Figure 2c. It can be seen that the characteristic impedance does not change much (<4 Ω when RF frequency changes from 10 to 70 GHz) with the varying frequency, which indicates a nearly perfect matching between the electrodes and terminal pads.

The fabrication process is briefly described below. First, a 200-nm-thick chromium (Cr) layer was deposited on the LNOI by magnetron sputtering. Then the Cr layer was patterned using femtosecond laser ablation and serves as the hard mask for the subsequent CMP etching process, which was carried out using a wafer polishing machine (PM6, Logitech, Lausanne, Switzerland). During the CMP process the LN thin film that is not protected by the Cr mask will be thinned and left with smooth sidewalls and thus form the LNOI waveguide circuits. After that, the residual Cr mask was removed using commercial Cr etchant (chromium etchant, Alfa Aesar GmbH, Haverhil, MA, USA) and the CPW electrodes were formed by evaporating a 500-nm-thick Cr/Au layer using electron beam evaporation (EBE) followed by standard lift-off process. Then a 100-nm-thick SiO_2_ and a 3-μm-thick SiON layer with a refractive index of 1.45 (@1550 nm) and 1.5 (@1550 nm), respectively, were deposited on the waveguides and electrodes using plasma enhanced chemical vapor deposition (PECVD). Finally, the SiON and SiO_2_ were etched using reactive ion etching (RIE) to expose the terminal pads and to form the SSCs before cleaving the chip.

## 3. Measurement of Mode Profile in the Waveguide

Figure 3a shows a photograph of a fabricated modulator chip. Fiber arrays were bonded on both ends of the chip using refractive index matching ultraviolet glue. The microscope image of the EO modulator is shown in Figure 3b. Figure 3c shows the SEM image of the cross section of the fabricated thin-film LN waveguide. Figure 3d shows the SEM image of the SSC, which can provide adiabatic conversion of the sub-micrometer mode area in the LN waveguide to micrometer mode area in the SiON waveguide. The mode field images captured by an infrared charged coupled device (CCD) are shown in Figure 3e,f for both the SiON waveguide and UHNA fiber (UHNA7, CORNING Inc., Corning, NY, USA), respectively. As can be seen from the images, the SSC offers favorable mode field overlap between the LN waveguide and the UHNA fiber.

To characterize the coupling loss between the SSC and the UHNA fiber, we coupled UHNA fibers with a straight LN waveguide of 1 mm and obtained a fiber-to-fiber insertion loss of ~2.0 dB. Since the propagation loss of such short waveguide can be negligible, the coupling loss between the SSC and the UHNA fibers can be estimated to be ~1.0 dB/facet. The fiber-to-fiber insertion loss of the fabricated EO modulator device was measured to be ~2.6 dB, including a coupling loss of 1.0 dB/facet and an on-chip propagation loss of ~0.6 dB. The coupling loss mainly results from the sidewall roughness of the dry etched SiON waveguides, which can be improve by optimizing the RIE process.

## 4. Performance Characterization of the Fabricated Device

The schematic measurement set-up for the broadband EO response characterization of the device under test (DUT) is shown in Figure 4. A laser beam at 1550 nm wavelength was sent into an inline fiber polarization controller (FPC) to generate the TE polarization mode as the light source of the DUT. The light was coupled into and out of the DUT using UHNA fibers. An RF signal with a frequency sweep continuously from 10 MHz up to 50 GHz was provided by a vector network analyzer (VNA, ZNA 50, Rohde & Schwarz GmbH & Co KG, Munich, Germany) which was applied on the DUT through a 50 GHz GSG probe (model 50A, GGB industries, Annecy, France). An additional probe was used to connect the DUT with an external 50 Ω terminator to reduce the RF signal reflection caused by impedance mismatching. The modulated light was converted to RF signal by a photodiode (PD, model1014, Newport Corporation, Santa Clara, CA, USA) and recorded by the VNA to analyze the S parameters.

To verify the calculation result of the RF field, we further measured the broadband electrical-electrical (EE) response of the electrodes on the fabricated modulator. Figure 5a shows the EE transmission (S21) and reflection (S11) parameter for the modulator with 7-mm-long arm length. The roll off of the S21 curve is 4.9 dB, implying an acceptable RF loss, since the 3 dB EO bandwidth can be estimated by the 6.41 dB EE bandwidth in the case of perfect velocity matching [18]. The S21 curve maintains below −30 dB, implying a good matching of the characteristic impedance Z_0_ between the CPW electrodes and the probes. The extracted Z_0_ together with the extracted RF group index verses the applied RF frequency are shown in Figure 5b, indicating that the RF group index is close to the optical group index with a difference below 0.05.

To characterize the static electro-optical (EO) property of the fabricated modulators, a 100 kHz triangle wave signal with a peak-to-peak velocity of 20 V was applied on the modulator. The EO responses of the 7-mm-long modulator is shown in Figure 5a,b, from which the *V_π_* was measured to be 3.1 V. The extinction ratio was measured to be −18 dB, thus the extracted *V_π_L* can be calculated as 2.16 V⋅cm. We then measured the broadband EO response of the fabricated modulators. The EO S21 measured as a function of the applied RF frequency from 10 MHz to 50 GHz is shown in Figure 6c. One can see that the actual 3-dB bandwidth of the modulator is beyond 50 GHz, which is limited by the performance of our measurement equipment.

## 5. Conclusions

In conclusion, we have shown that high performance EO modulators featuring a low fiber-to-fiber insertion loss below 3 dB, a low *V_π_* of 3.1 V and a high EO bandwidth over 50 GHz have been fabricated using the PLACE technique. The performance of the device achieved in this work is almost on par with the best performance devices demonstrated. However, we would like to stress that using our fabrication technology the performance is highly stable and reproducible. Specifically, thanks to the verified competitive production rate (i.e., the mask patterning step by femtosecond laser micromachining takes only 3 min for each modulator, which corresponds to annual production yield of 150,000 pcs/y) we have achieved, the fabrication technique reported by us in the current manuscript is of critical importance for the widespread application of high performance EO modulators with low insertion loss. All of these advantages not only greatly benefit the commercialization of TFLN EO modulators but also pave the way for the future development of large scale photonic integrated devices, such as large-scale low-loss cascaded modulators, artificial neural networks and so on.

## Figures and Tables

**Figure 1 micromachines-13-00378-f001:**
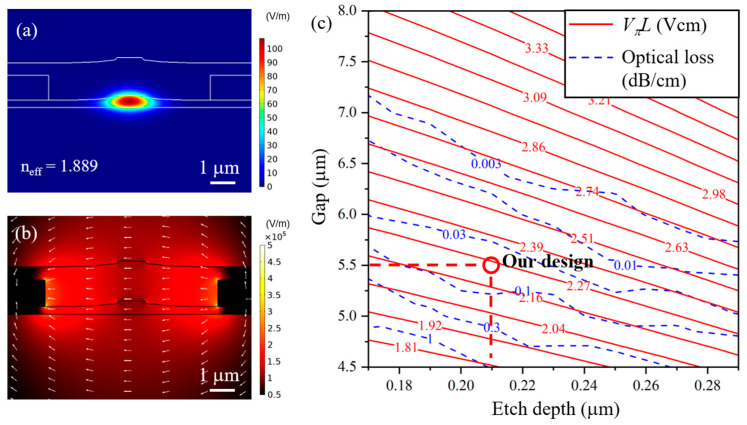
(**a**) Simulated optical field profile of the fundamental transverse electric (TE) mode in the lithium niobate (LN) ridge waveguide. (**b**) Simulated static electric field when applying a 1 V voltage between the electrodes. (**c**) Contour map of both *V_π_L* and optical loss versus different gaps and etching depths.

**Figure 2 micromachines-13-00378-f002:**
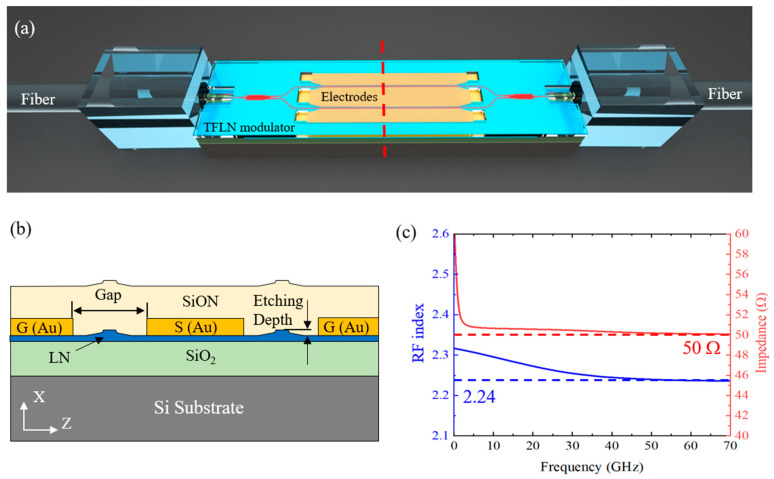
(**a**) Picture of the thin-film lithium niobate (TFLN) modulator schematic, ultra-high numerical aperture (UHNA) fiber arrays are placed on both sides to couple light into and out of the chip. (**b**) Cross section of the modulation area. (**c**) The simulated RF group index and characteristic impedance of the modulator.

**Figure 3 micromachines-13-00378-f003:**
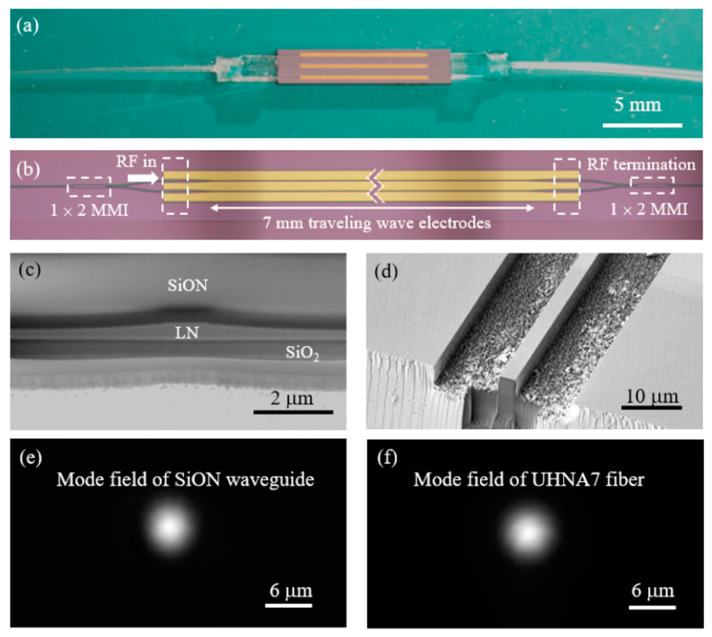
(**a**) Photograph of a fabricated electro-optic (EO) modulator bonded with fiber arrays. (**b**) The microscope image of the EO modulator part of the chip. (**c**) The SEM image of the cross section of the fabricated thin-film LN waveguide. (**d**) The SEM image of the SiON waveguide. (**e**) The mode field profile of the SiON waveguide. (**f**) The mode field profile of the commercial UHNA7 fiber.

**Figure 4 micromachines-13-00378-f004:**
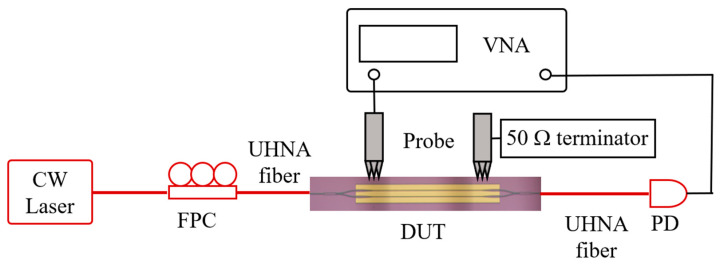
The schematic view of the measurement set-up for broadband EO response characterization. EDFA, erbium-doped fiber amplifier; DUT, device under test; FPC, fiber polarization controller; PD, photodiode; VNA, vector network analyzer.

**Figure 5 micromachines-13-00378-f005:**
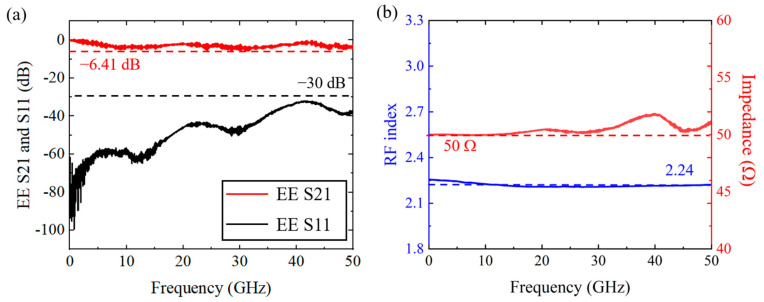
(**a**) The EE transmission (S21) and reflection (S11) parameter for the 7-mm-long modulator. (**b**) The extracted RF group index curve and characteristic impedance curve of the coplanar waveguide (CPW) electrodes.

**Figure 6 micromachines-13-00378-f006:**
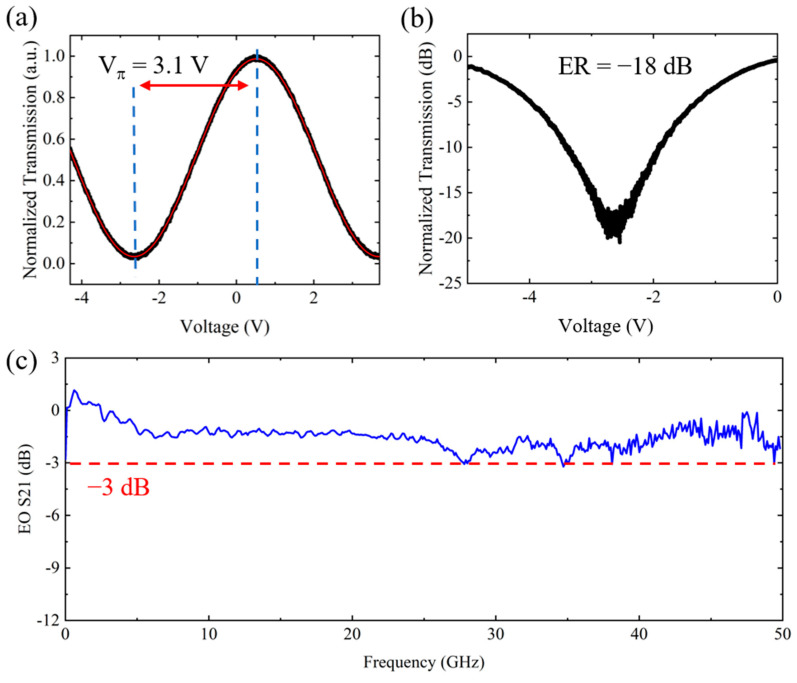
(**a**) The static EO response of the modulator. (**b**) The static EO response of the modulator under logarithmic scale. (**c**) The EO S21 curve up to 50 GHz of the modulator.

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
