# Peer review of "High-Production-Rate Fabrication of Low-Loss Lithium Niobate Electro-Optic Modulators Using Photolithography Assisted Chemo-Mechanical Etching (PLACE)"

_micromachines, 2022, doi:10.3390/mi13030378_

Round 1
Reviewer 1 Report
In the present manuscript titled "High-production-rate fabrication of low-loss lithium niobate electro-optic modulators using photolithography assisted chemo-mechanical etching (PLACE)", the authors have demonstrated good quality thin-film lithium niobate modulators that are desirable for integrated photonic chips. While the topic of the paper is of high interest and encouraging results, the manuscript in the present form provides insufficient information on the fabrication process and lacks the scientific rigor required. Thus, I recommend that authors revise the manuscript before it can be accepted for publication. Some of the specific comments and concerns are as followed -
- During the introduction, the authors have simply mentioned similar material platforms (lines 35-36) without describing their benefits and drawbacks. Additionally, "plasmonic" is not a material platform but a way of implementation, could authors elaborate on it?
- The authors have not sufficiently critiqued and referenced the existing fabrication methods to justify the approach they have taken. Performances of the devices fabricated with them and their throughput are both needed to be described to set a benchmark for the PLACE approach authors have taken.
- The authors have not sufficiently described what the PLACE technique in fact is and how does it distinguish from other lithography and direct-write techniques? (eg. UV lithography is in fact photolithography, how is PLACE different then? Electron-beam writing term is ambiguous - is it electron beam lithography followed by reactive ion etching or is it electron beam induced direct-write or direct-etch)
- The authors have not given any details of the fabrication process they have implemented to fabricate these TFLN modulators. It is important that the readers should be able to reproduce the author's results.
- Line 67 - authors claim that they have simulated waveguide fabricated by PLACE technology, which is not true. The simulations merely take into account the geometry and material properties the authors intend to fabricate. SImulations do not take into account the process effects in the simulations.
- Line 69 - "Figure1b" is wrongly referenced as "Figure2b"
- Authors should specify how they have set their simulations (which packages, what are standard materials properties, and boundary conditions they have used)
- Line 89 - authors have not specified any details of how the SiON layer is deposited, which is critical to the material properties.
- Line 100 - "matches nicely" could authors elaborate?
- LIne 103 - authors claim "the impedance does not change much" but the impedance does go up asymptotically near the low-frequency region.
- As specified by authors later in line 170, they measure over a frequency range of 10 MHz to 50 GHz. In order to appropriately emphasize the performance over this range, authors should consider using a logarithmic scale for the frequency in all FIgure 2c, Figure 5b and Figure 6c
- Figure 2a - could authors label the different parts of the modulator e.g. modulator, fibers, transmission line, electrodes etc.
- Figure 2b - could authors also show which distance is considered as "etch depth"
- Figure 3c - could authors label various materials layers?
- FIgure3d caption - is it SiON waveguide or LN waveguide? Also, could the authors discuss the surface roughness seen in the etched regions and how would that impact the device performance?
- Line 153 - what do authors mean by "roll-off" and how is it measured?
- Figure 5a - COuld authors add legends, readers wouldn't know which is which curve is S21 and which is S11
- Figure 6b - What is the X-axis represent? How is extinction ratio defined?
- Figure 6c has no caption
- The conclusions need rewriting to appropriately summarize the key takeaways of the paper. The conclusions section is not the place for process details, in fact, there should be a separate experimental section in the paper.
Author Response
We thank the editor and all the reviewers for their constructive remarks and useful suggestions, which helps us significantly raised the quality of the manuscript. Each suggested revision and comment have been addressed point by point as below.
General Comments:
In the present manuscript titled "High-production-rate fabrication of low-loss lithium niobate electro-optic modulators using photolithography assisted chemo-mechanical etching (PLACE)", the authors have demonstrated good quality thin-film lithium niobate modulators that are desirable for integrated photonic chips. While the topic of the paper is of high interest and encouraging results, the manuscript in the present form provides insufficient information on the fabrication process and lacks the scientific rigor required. Thus, I recommend that authors revise the manuscript before it can be accepted for publication. Some of the specific comments and concerns are as followed -
- Comment: During the introduction, the authors have simply mentioned similar material platforms (lines 35-36) without describing their benefits and drawbacks. Additionally, "plasmonic" is not a material platform but a way of implementation, could authors elaborate on it?
Reply: We gratefully appreciate for your valuable suggestion; we have added the description about the benefits and drawbacks of different material platforms in the revised manuscript (P.1 L.40 “Although these platforms separately offer the advantages of compact footprints (Si, plasmonics) and low drive voltages (InP, polymer), drawbacks including large Vπ (Si), high optical loss (plasmonics), nonlinear response (InP), or stability problem (polymer) still exists in modulators based on these material platforms.”). Besides, as far as we know, “plasmonics” often be classified as a material platform, actually, reference [14] and [24] also describe plasmonics as a material platform.
- Comment: The authors have not sufficiently critiqued and referenced the existing fabrication methods to justify the approach they have taken. Performances of the devices fabricated with them and their throughput are both needed to be described to set a benchmark for the PLACE approach authors have taken.
Reply: Thank you for your rigorous comment, we have added citation and description about the existing fabrication methods (P.2 L.53 “Electron beam lithography followed by reactive ion etching technology can provide sufficiently high fabrication precision whilst suffers from a relatively low production yield (more than 8 days for 6-inch wafer exposure [16]). Ultraviolet (UV) lithography technology can provide high fabrication efficiency, whilst uncertainty still exists in terms of uniformity of wafer-scale production and propagation loss (over 0.1 dB/cm) induced by the sidewall roughness [18,21].”).
- Comment: The authors have not sufficiently described what the PLACE technique in fact is and how does it distinguish from other lithography and direct-write techniques? (eg. UV lithography is in fact photolithography, how is PLACE different then? Electron-beam writing term is ambiguous - is it electron beam lithography followed by reactive ion etching or is it electron beam induced direct-write or direct-etch)
Reply: We agree with the reviewer, we have added the description of the PLACE technique in our revised manuscript (P.2 L.61 “Utilizing the femtosecond direct writing technology for Cr hard mask patterning and followed by chemo-mechanical polish (CMP) technology for waveguide etching, the PLACE technology support large device footprint (footprint size over 12-inch), high fabrication uniformity, and competitive production rate (take only ~3 min to mask patterning for each modulator) simultaneously thanks to the high average power femtosecond laser and high-speed large-motion-range position stages. PLACE also provides extreme low device optical loss thanks to the extreme smooth waveguide sidewall (roughness less than 0.1 nm) produced by the CMP process.”). We also modified the statement about the electro-beam lithography technology to “Electron beam lithography followed by reactive ion etching technology” in P.2 L.53.
- Comment: The authors have not given any details of the fabrication process they have implemented to fabricate these TFLN modulators. It is important that the readers should be able to reproduce the author's results.
Reply: As for the referee’s concern, we have added paragraph to describe the details about the fabrication process in our revised manuscript (P.3 L.130 “The fabrication process is briefly descripted as below. First, a 200-nm thick chromium (Cr) was deposited on the LNOI by magnetron sputtering. Then the Cr layer were patterned using femtosecond laser ablation and serves as the hard mask for subsequent CMP etching process, which was carried out using a wafer polishing machine (Logitech PM6), during the CMP process the LN thin film that is not protected by the Cr mask will be thinned and left with smooth sidewalls and thus form the LNOI waveguide circuits. After that, the residual Cr mask was removed using commercial Cr etchant and the CPW electrodes was formed by evaporating a 500-nm-thick Cr/Au layer using electron beam evaporation (EBE) followed by standard lift-off process. Then a 100-nm-thick SiO2 and a 3-mm-thick SiON layer with a refractive index of 1.45 (@1550 nm) and 1.5 (@1550 nm) separately were deposited on the waveguides and electrodes using plasma enhanced chemical vapor deposition (PECVD). Finally, the SiON and SiO2 were etched using reactive ion etching (RIE) to expose the terminal pads and to form the SSCs before cleaving the chip.”).
- Comment: Line 67 - authors claim that they have simulated waveguide fabricated by PLACE technology, which is not true. The simulations merely take into account the geometry and material properties the authors intend to fabricate. Simulations do not take into account the process effects in the simulations.
Reply: We understand referee’s concern, actually the geometries used in the simulations were extract out from the SEM image of the cross sections of TFLN waveguides fabricated using PLACE technology under different fabrication conditions, the process effects were taken into account. We have modified the description about the simulation to avoid misunderstanding in our revised manuscript (P.2 L.84 “the geometries used in the simulation were extracted from the SEM images of the cross sections of waveguides fabricated using PLACE technology.”).
- Comment: Line 69 - "Figure1b" is wrongly referenced as "Figure2b"
Reply: Thank you for your careful check. We have the statement to “Figure 1b” in P.2 L.87.
- Comment: Authors should specify how they have set their simulations (which packages, what are standard materials properties, and boundary conditions they have used)
Reply: Thank you for your comment, we have added the details about the simulations in the revised manuscript. (P.2 L.83 “using the COMSOL Multiphysics under perfect matched layer (PML) boundary condition” P.3 L.119 “using HFSS, under absorbing condition”)
- Comment: Line 89 - authors have not specified any details of how the SiON layer is deposited, which is critical to the material properties.
Reply: Thank you for your comment, the SiON layer is deposited using PECVD technology, we have added the details about the simulations in the revised manuscript. (P.4 L.141 “using plasma enhanced chemical vapor deposition (PECVD).”)
- Comment: Line 100 - "matches nicely" could authors elaborate?
Reply: Thank you for your comment, we have modified the statement about the simulated group index matching in the revised manuscript. (P.3 L.122 “mismatching below 0.02 when RF frequency from 40 GHz to 70 GHz”)
- Comment: Line 103 authors claim "the impedance does not change much" but the impedance does go up asymptotically near the low-frequency region.
Reply: Actually, the mismatching of the impedance only affect the performance of the modulator near the high-frequency region. As for the referee’s concern, we added the description of “<4 W when RF frequency change from 10 GHz to 70 GHz” in P.3 L.126 in the revised manuscript.
- Comment: As specified by authors later in line 170, they measure over a frequency range of 10 MHz to 50 GHz. In order to appropriately emphasize the performance over this range, authors should consider using a logarithmic scale for the frequency in all Figure 2c, Figure 5b and Figure 6c
Reply: Thank you for your kind suggestion, using a logarithmic scale for frequency will emphasize the low-frequency performance of the modulator, however, the high-frequency performance of the modulator is more important for a modulator. Besides, if we are not wrong, most reported studies including references [11-24] about broad bandwidth modulators tends to using leaner scale rather than logarithmic scale for the frequency. So, we decide to keep using linear scale for the frequency in our revised manuscript.
- Comment: Figure 2a - could authors label the different parts of the modulator e.g. modulator, fibers, transmission line, electrodes etc.
Reply: Thank you for your kind suggestion, we have labeled the different parts of the modulator in Figure 2a in the revised manuscript. (P.4 L.144)
- Comment: Figure 2b - could authors also show which distance is considered as "etch depth"
Reply: Thank you for your kind suggestion, we have labeled the etch depth in Figure 2b in the revised manuscript. (P.4 L.144)
- Comment: Figure 3c - could authors label various materials layers?
Reply: Thank you for your comment, we have labeled various materials layers 3c in the revised manuscript. (P.7 L.168)
- Comment: FIgure3d caption - is it SiON waveguide or LN waveguide? Also, could the authors discuss the surface roughness seen in the etched regions and how would that impact the device performance?
Reply: As for referee’s concern, Figure 3d shows the SEM image of the SiON waveguide. coupling loss of the SSCs will increase because of the roughness in the etched regions, which results from the unoptimized etching process for the SiON layer. We add the related discussions in our revised manuscript. (P.5 L.165 “The coupling loss mainly results from the sidewall roughness of the dry etched SiON waveguides, which can be improve by optimizing the RIE process.”)
- Comment: Line 153 - what do authors mean by "roll-off" and how is it measured?
Reply: As for referee’s concern, “roll-off” means the lowest value of the S21 when the applied frequency changing from 10 MHz to 50 GHz.
- Comment: Figure 5a - Could authors add legends, readers wouldn't know which is which curve is S21 and which is S11
Reply: Thank you for your kind suggestion, we have added the legends of S21 and S11 in Figure 5a in the revised manuscript. (P.9 L.201)
- Comment: Figure 6b - What is the X-axis represent? How is extinction ratio defined?
Reply: Thank you for your careful check, we have added the title of X-axis a in the revised manuscript. (P.10 L.215) The extinction ratio is defined as the ratio of the minimum light intensity and the maximum light intensity.
- Comment: Figure 6c has no caption
Reply: Thank you for your careful check, we have added the caption in figure 6c in the revised manuscript. (P.10 L.215)
- Comment: The conclusions need rewriting to appropriately summarize the key takeaways of the paper. The conclusions section is not the place for process details, in fact, there should be a separate experimental section in the paper.
Reply: Thanks to the reviewers for the nice suggestion. We have rewrite the conclusions section in the revised section. (P.11 L.227 “In conclusion, we have shown that high performance EO modulators featuring a low fiber-to-fiber insertion loss below 3 dB, a low Vp of 3.1 V, and a high EO bandwidth over 50 GHz have been fabricated using the PLACE technique. The performance of the device achieved in this work is almost on par with the best performance devices demonstrated. However, we would like to stress that using our fabrication technology the performance is highly stable and reproducible. Specifically, thanks to the verified competitive production rate (i.e., the mask patterning step by femtosecond laser micromachining takes only 3-min for each modulator, which corresponds to annual production yield of 150, 000 pcs/year) as we have achieved, the fabrication technique reported by us in the current manuscript is of critical importance for the widespread application of high performance EO modulators with low insertion loss. All of these advantages not only benefit greatly the commercialization of TFLN EO modulators but also pave the way for the future development of large scale photonic integrated devices such as large-scale low-loss cascaded modulators, artificial neural networks and so on.”)

Reviewer 2 Report
The paper describes the performance of a wide bandwidth thin film lithium niobate optical MZ modulator.
As also stated by the authors, this material has already been widely investigated in recent years, even by the authors themselves, as reported in the appropriate self citation references.
Therefore, the motivation of the research must be clearly stated. Is it a record performance, or any other reason that makes this paper interesting?
I am not contesting the value of the paper here, but I suggest the authors to state already in the abstract (not declared here) and then make it very clear in the Conclusions that the main interesting feat here is speed of fabrication thanks to the chosen and developed fabrication process.
If I am not wrong. Otherwise, it means the authors should make it even more clear. ;-)
Because as far as I could find in the literature, the performance of the device described in this paper is not showing a record, though it definitely is almost on par with the best performance devices demonstrated. Perhaps, also this aspect should be better enlightened and clarified, in the discussion of the characteristics of the device object of the paper.
The paper largely lacks the description of the fabrication parameters. Large space is devoted to the description of the design simulation and choices, but then very few details about the fabrication techniques are give.
I understand that these can (perhaps, since no direct references or confirmation on this aspect are given) be deducted by the self-cited articles, but nevertheless an article needs to be complete on his own, in case the secondary sources are not available for any reason at the moment of reading.
Why was the use of an optical amplifier necessary during the characterisation?
References are extensive, but I suggest to add following papers for completeness and coherence:
- High-performance coherent optical modulators
based on thin-film lithium niobate platform, DOI:10.1038/s41467-020-17806-0 (if I am not wrong better performance than device in this paper). - Thin film lithium niobate electro-optic
modulator with terahertz operating bandwidth, DOI:10.1364/OE.26.014810 (large bandwidth, though large losses).
Author Response
Reviewer 2
General Comments:
The paper describes the performance of a wide bandwidth thin film lithium niobate optical MZ modulator.
As also stated by the authors, this material has already been widely investigated in recent years, even by the authors themselves, as reported in the appropriate self citation references.
Therefore, the motivation of the research must be clearly stated. Is it a record performance, or any other reason that makes this paper interesting?
I am not contesting the value of the paper here, but I suggest the authors to state already in the abstract (not declared here) and then make it very clear in the Conclusions that the main interesting feat here is speed of fabrication thanks to the chosen and developed fabrication process.
If I am not wrong. Otherwise, it means the authors should make it even more clear. ;-)
Because as far as I could find in the literature, the performance of the device described in this paper is not showing a record, though it definitely is almost on par with the best performance devices demonstrated. Perhaps, also this aspect should be better enlightened and clarified, in the discussion of the characteristics of the device object of the paper.
Reply: Thanks to the reviewer for the nice comment, yes we should make a clearer statement on the motivation of this work, we have rewrite the conclusions section in the revised section as below.
“In conclusion, we have shown that high performance EO modulators featuring a low fiber-to-fiber insertion loss below 3 dB, a low Vp of 3.1 V, and a high EO bandwidth over 50 GHz have been fabricated using the PLACE technique. The performance of the device achieved in this work is almost on par with the best performance devices demonstrated. However, we would like to stress that using our fabrication technology the performance is highly stable and reproducible. Specifically, thanks to the verified competitive production rate (i.e., the mask patterning step by femtosecond laser micromachining takes only 3-min for each modulator, which corresponds to annual production yield of 150, 000 pcs/year) as we have achieved, the fabrication technique reported by us in the current manuscript is of critical importance for the widespread application of high performance EO modulators with low insertion loss. All of these advantages not only benefit greatly the commercialization of TFLN EO modulators but also pave the way for the future development of large scale photonic integrated devices such as large-scale low-loss cascaded modulators, artificial neural networks and so on.”
We also added the statement of “The PLACE technology supports large footprint, high fabrication uniformity, competitive production rate and extreme low device optical loss simultaneously, our result shows promising potential for developing high-performance large-scale low-loss photonic integrated devices.”
The paper largely lacks the description of the fabrication parameters. Large space is devoted to the description of the design simulation and choices, but then very few details about the fabrication techniques are give.
I understand that these can (perhaps, since no direct references or confirmation on this aspect are given) be deducted by the self-cited articles, but nevertheless an article needs to be complete on his own, in case the secondary sources are not available for any reason at the moment of reading.
Reply: We gratefully appreciate for your valuable suggestion; we have added paragraph to describe the details about the fabrication process in our revised manuscript (P.3 L.123 “The fabrication process is briefly descripted as below. First, a 200-nm thick chromium (Cr) was deposited on the LNOI by magnetron sputtering. Then the Cr layer were patterned using femtosecond laser ablation and serves as the hard mask for subsequent CMP etching process, which was carried out using a wafer polishing machine (Logitech PM6), during the CMP process the LN thin film that is not protected by the Cr mask will be thinned and left with smooth sidewalls and thus form the LNOI waveguide circuits. After that, the residual Cr mask was removed using commercial Cr etchant and the CPW electrodes was formed by evaporating a 500-nm-thick Cr/Au layer using electron beam evaporation (EBE) followed by standard lift-off process. Then a 100-nm-thick SiO2 and a 3-mm-thick SiON layer with a refractive index of 1.45 (@1550 nm) and 1.5 (@1550 nm) separately were deposited on the waveguides and electrodes using plasma enhanced chemical vapor deposition (PECVD). Finally, the SiON and SiO2 were etched using reactive ion etching (RIE) to expose the terminal pads and to form the SSCs before cleaving the chip.”).
Why was the use of an optical amplifier necessary during the characterization?
Reply: EDFA is unnecessary since the insertion loss of the modulator is low enough, we used it in our previous work and forgot to remove it in our manuscript, we have removed it in the revised manuscript.
References are extensive, but I suggest to add following papers for completeness and coherence:
High-performance coherent optical modulators
based on thin-film lithium niobate platform, DOI:10.1038/s41467-020-17806-0 (if I am not wrong better performance than device in this paper).
Thin film lithium niobate electro-optic
modulator with terahertz operating bandwidth, DOI:10.1364/OE.26.014810 (large bandwidth, though large losses).
Reply: We agree with the referee, we have added the references as referee’s suggestion.
